# Disruptive Effects of Two Curcuminoids (Demethoxycurcumin and Bisdemethoxycurcumin) on the Larval Development of *Drosophila melanogaster*

**DOI:** 10.3390/insects14120959

**Published:** 2023-12-18

**Authors:** Jun-Hyoung Jeon, Seon-Ah Jeong, Doo-Sang Park, Hong-Hyun Park, Sang-Woon Shin, Hyun-Woo Oh

**Affiliations:** 1Biological Resource Center, Korea Research Institute of Bioscience and Biotechnology, Jeongeup 56212, Republic of Korea; wjs258@kribb.re.kr (J.-H.J.); jeong420@kribb.re.kr (S.-A.J.); dspark@kribb.re.kr (D.-S.P.); 2Crop Protection Division, National Academy of Agricultural Science, Rural Development Administration, Wanju 55365, Republic of Korea; honghyunpark@korea.kr; 3Core Facility Management Center, Korea Research Institute of Bioscience and Biotechnology, Daejeon 34141, Republic of Korea

**Keywords:** juvenile hormone, methoprene-tolerant, diterpene, juvenile hormone disruptor, curcuminoids, demethoxycurcumin, bisdemethoxycurcumin

## Abstract

**Simple Summary:**

Several plant species have diterpene compounds with juvenile hormone (JH) disruptor activity in insects. Demethoxycurcumin (DMC) and bisdemethoxycurcumin (BDMC) are two curcuminoid components of turmeric, interfering with the JH-mediated formation of the JH receptor complex. In vitro, DMC and BDMC also inhibit larval and pupal development in *Drosophila melanogaster*. The results suggest that both DMC and BDMC are JH disruptors that affect insect growth and development by regulating JH-mediated gene expression.

**Abstract:**

Juvenile hormones (JHs) play a central role in insect development, reproduction, and various physiological functions. Curcuminoids generally exhibit a wide range of biological activities, such as antioxidant, anti-inflammatory, antibacterial, and insecticidal, and they exhibit insect growth inhibitory effects. However, research on insecticidal properties of curcuminoids has been limited. Moreover, to the best of our knowledge, studies on JHs of insects and curcuminoids are lacking. Therefore, this study aimed to identify the substances that act as JH disruptors (JHDs) from edible plants. Demethoxycurcumin (DMC) and bisdemethoxycurcumin (BDMC), two curcuminoids from the turmeric plant *Curcuma longa* L. inhibited the formation of a methoprene-tolerant (Met)–Taiman (Tai) heterodimer complex in *Drosophila melanogaster*, as shown through in vitro yeast two-hybrid assays. An artificial diet containing 1% (*w*/*v*) DMC or BDMC significantly reduced the number of *D. melanogaster* larvae in a concentration-dependent manner; larval development was disrupted, preventing the progression of larvae to pupal stages, resulting in an absence of adults. Building on the results obtained in this study on curcuminoids, researchers can use our study as a reference to develop eco-friendly pesticides.

## 1. Introduction

While chemical insecticides with diverse insecticidal effects have long been utilized to control major pests, issues including ecosystem degradation, environmental contamination, and the emergence of resistance in target pests due to the improper use and overuse of these insecticides have arisen [1]. Therefore, eco-friendly insecticides that exhibit effective insecticidal effects without considerably affecting the environment have been actively researched, and several have been characterized from different types of plant extracts [2]. Plants use secondary metabolites including nicotine, rotenone, lianas, sabadala, pyrethrum, neem, and turmeric as natural defenses against insect pests [3]. Turmeric is derived from the root of *Curcuma longa* L., a perennial herb belonging to the Zingiberaceae family [4,5], and contains approximately 60–70% carbohydrates, 6–8% proteins, 5–10% fat, 3–7% minerals, and 6–13% moisture [5,6,7,8]. Within turmeric, there are more than 50 structurally related compounds known as curcuminoids, which make up approximately 3–5% of the total composition. The main curcuminoids include three commercially available compounds: curcumin, demethoxycurcumin (DMC), and bisdemethoxycurcumin (BDMC) [6,9]. Curcuminoids are well known for their various pharmacological properties, including antidepressant, antioxidant, anti-inflammatory, hepatoprotective, antidiabetic, anticancer, and antibacterial effects [7,10,11,12,13,14]. However, research on the insecticidal properties of curcuminoids has been limited. Those research studies conducted on insecticidal properties explored the efficacy of curcuminoids, including ovicidal activity on *Helicoverpa armigera* eggs [15]. In particular, curcuminoids I, II, and III have been shown to inhibit the P-glycoprotein ATPase in *H. armigera* [16] and display insecticidal activity against *Aedes aegypti* [17], while other curcuminoids have shown high efficacy against *Aedes albopictus* and *Culex pipiens* larvae [18]. In particular, curcumin, demethoxycurcumin, a curcumin–BF2 complex, and a monocarbonyl tetramethoxy–curcumin derivative have exhibited remarkable larvicidal activity, indicating their potential as alternative agents for mosquito control strategies [18]. Curcumin was also found to increase mortality in the early larval stages of *C. pipiens* through acetylcholine esterase 1 (AchE1) inhibition [19]. Additionally, research has explored the anti-insect effects of Curcuma essential oil, such as feeding inhibition, oviposition deterrence, and reproductive inhibition [20]. However, to our knowledge, no studies on the juvenile hormones (JHs) of insects related to curcuminoids have been conducted.

The intricate interplay between JH and ecdysone in insect development is a finely tuned process. JH, synthesized in the corpus allatum, interacts with ecdysone to maintain larval development during molts [21]. Beyond its role in molting, JH plays a multifaceted role in crucial physiological functions, encompassing reproduction, developmental regulation, pheromone production, and caste differentiation, particularly in social insects [22]. Methoprene-tolerant (Met) was identified as a JH receptor in mutant studies of *Drosophila melanogaster*. This accelerated research on JH signaling [23]. Derived from the germ cell-expressed gene, Met is a member of the basic helix–loop–helix (bHLH) Per-Arnt-Sim (PAS) family [24,25,26]. Like other members of the bHLH-PAS transcription factor family, Met requires the formation of a heterodimeric partnership with other bHLH-PAS proteins for its activation [27]. JHs act through a receptor complex consisting of Met and a steroid receptor coactivator (SRC); Taiman in *D. melanogaster*, or βFtz-F1 Interacting Steroid Receptor Coactivator FISC in *A. aegypti* to induce the transcription of specific genes [28]. Met heterodimerizes with an SRC, binds to JH with high affinity, and activates transcription [23,24,25,26]. To investigate the impact of these interactions on the JH signaling pathway, in vitro yeast two-hybrid assay systems have been employed, utilizing the JH-dependent heterodimer-binding properties of Met and SRC. These experimental approaches have facilitated the exploration of JH Disruptor (JHD) activities of various plant extracts, not only in *A. aegypti* [29], but also in the Indian meal moth, *Plodia interpunctella* [30]. Furthermore, an in vitro assay system capable of quantifying the disruption activity of plant extracts and diterpenes on JH-mediated Met–Taiman (Tai) heterodimer formation in *D. melanogaster* was developed [31]. The outcomes of these studies collectively underscore the prevalence of JHD diterpenes in the plant kingdom, emphasizing their propensity to interfere with JH-mediated endocrine regulation in insects [32].

In this study, we carefully evaluated the JHD activity of 30 herbal medicines and edible plant extracts, building on previous findings that highlighted their potent JHD effects [31]. Both the plant extracts from *C. longa* L. and two curcuminoids, DMC and BDMC, showed strong JHD activity in the in vitro assay system of *D. melanogaster* Met–Tai. DMC and BDMC also blocked larval/pupal development, preventing the formation of pupae and the emergence of adults.

## 2. Materials and Methods

### 2.1. Chemicals and Insects

JH III and methoprene, integral to our research, were acquired from Sigma-Aldrich (St. Louis, MO, USA). Additionally, three curcuminoids—curcumin, DMC, and BDMC—were procured from Aladdin (Pudong, Shanghai, China). The plant diterpene methyl lucidone (LE3G) was extracted from *Lindera erythrocarpa*, following the methodology outlined in a previous study [29]. We sourced plant extracts, specifically methanol extracts, from 30 distinct edible plant species from the Korean Plant Extracts Bank (Daejeon, Republic of Korea). Each reagent was prepared as a stock solution with a concentration of 20 mg/mL, using either dimethyl sulfoxide (DMSO) or methanol as the solvent. For *D. melanogaster* culture, we used instant fruit fly medium from Hansol Tech (Seoul, Republic of Korea). This comprehensive set of reagents and growth media served as the basic elements of our experimental framework, ensuring reliability and precision in our JHD activity analyses.

### 2.2. Yeast β–Galactosidase Assay

The yeast two-hybrid binding assay, utilizing a quantitative β–galactosidase assay, was conducted with the Y187 strain transformed by the *D. melanogaster* JH receptor and its interacting partner, Met–Tai, following the established protocol outlined in a prior study [31]. The Y187 strain, which was modified according to the protocol specified in the previous study [31], was grown in culture for the screening of JHD activity. A volume of 100 µL of cultured yeast cells with an optical density of 600 nm (OD_600_), ranging from 0.2 to 0.3, were exposed to a combination of 0.1 ppm JH III and 10 ppm individual plant extracts. This exposure was performed in 96-well plates, with each plate including a positive control comprising 0.1 ppm JH III with 10 ppm methyl lucidone, and a negative control involving 0.1 ppm JH III with the control solvent (DMSO). Following a 3 h incubation period, the cells were subjected to an assay for OD_420_, providing a quantitative measure of *β*–galactosidase activity. The OD_420_ values were then normalized to an arbitrary unit of JHD. Methyl lucidone served as a positive control owing to its strong interference with JH III-mediated Met–Tai binding in the *D. melanogaster β*–galactosidase assay systems tested [31]. A single arbitrary unit of JHD activity was determined based on the binding interference observed with 10 ppm methyl lucidone. In order to evaluate the specific JHD activity of each plant extract, the mean value of the duplicate experiments was calculated in arbitrary units according to the following formula:(1)$$A=\frac{{OD}_{420}\mathrm{Control}-{OD}_{420}PE}{{OD}_{420}\mathrm{Control}-{OD}_{420}ML10}$$

Here, A is the activity of the JHD, adjusted to A = 0 if A < 0. This adjustment ensures that negative values do not contribute to the specific JHD activity calculation. *OD*_420_
*Control* is the absorbance of yeast cells treated with JH III at 0.1 ppm; *OD*_420_
*PE* is the absorbance of yeast cells treated with 0.1 ppm of JH III and 10 ppm of each plant extract; and *OD*_420_
*ML*10 is the absorbance of cells treated with 0.1 ppm of JH III and 10 ppm of methyl lucidone.

The formula essentially compares the observed effects of the plant extract with those of the positive and negative controls, providing a standardized measure of the specific JHD activity that ensures accurate and meaningful comparisons across different experimental conditions.

### 2.3. Bioassay

The experimental design commenced by introducing 20 male and 20 female flies separately into individual vials. Each vial was then subjected to a specific dietary regimen. The treatments included 1% (*w*/*v*) concentrations of JHD (namely, LE3G, DMC, and BDMC), 0.05% (*w*/*v*) of the Juvenile Hormone Analog (JHA) methoprene, and a control group treated with 1% (*w*/*v*) ethanol. All of these compounds were thoroughly incorporated into 3 g of artificial feed. Each JHD (LE3G, DMC, and BDMC) was first dissolved in ethanol to a concentration of 20 mg/mL (100%). The 100% solution was then further diluted with ethanol to obtain a 10-fold solution corresponding to the desired JHD concentration. For example, to formulate a diet containing 1% JHD, one volume of the 10% stock solution was mixed with nine volumes of *Drosophila* diet to obtain a 1% solution. To ensure the proper solidification of the diet and the subsequent evaporation of ethanol, the mixed diets were allowed to stand undisturbed at room temperature overnight. Subsequently, the vials containing the experimental diets were maintained at a controlled breeding temperature of 25 °C. Following a one-day period of oviposition, during which each vial accumulated approximately 150–250 eggs, the adult flies were removed from the vials. After an additional 3–4 days, second-instar larvae were collected from each vial, forming the basis for subsequent analyses.

### 2.4. RNA Extraction, Primers, and Quantitative Real-Time Polymerase Chain Reaction Analysis

Total RNA was isolated from second-instar larvae using the RNeasy isolation kit (Qiagen, Hilden, Germany). Subsequently, cDNA was synthesized for a quantitative polymerase chain reaction (qPCR) with the iScript cDNA Synthesis Kit (Bio-Rad, Hercules, CA, USA) using 1 μg of RNA, and quantified with a NanoDrop ND-1000 Spectrophotometer (Thermo Scientific, Waltham, MA, USA). The qRT-PCR primers were designed through NCBI Primer-BLAST (https://www.ncbi.nlm.nih.gov/tools/primer-blast/, accessed on 7 June 2023); the primer sequences are listed in Table 1. For qRT-PCR, the SsoAdvanced Universal SYBR Green Supermix (Bio-Rad) was employed in 96-well plates on a CFX Connect Real-Time Polymerase Chain Reaction System (Bio-Rad). The thermal cycling program consisted of an initial denaturation at 95 °C for 30 s, followed by 40 cycles at 95 °C for 10 s and 60 °C for 30 s. A final melting curve analysis was performed at 95 °C for 5 s and 65 °C for 0.5 s. Validation of amplification efficiency and specificity was performed using CFX Maestro Software (Bio-Rad). To normalize the signal intensity of *Kr-h1*, it was compared to that of rp49. All RNA extractions and qRT-PCR analyses were conducted in triplicate, and the means of each sample were compared.

### 2.5. Data Analysis

The significance of mean differences between treatment and control groups was assessed by ANOVA with multiple comparisons followed by the appropriate Tukey HSD test. *p* < 0.05 was considered statistically significant. Statistical analyses and figure preparations were conducted using GraphPad Prism 7 software.

## 3. Results

### 3.1. JHD Activity of Plant Extracts

In order to evaluate the JHD activities of different edible plant species on the formation of Met–Tai heterodimers in *D. melanogaster*, 30 plant extracts, previously identified in a comprehensive study and mainly used as herbal medicines and edible plant extracts, were selected for investigation [32]. The selected plant extracts were tested for JHD activity in *D. melanogaster* using yeast two-hybrid *β*–galactosidase assays (Table 2). Among the extracts tested, 14 showed substantial JHD activity, with the highest activity observed in *C. longa* L. Notably, the curcuminoids present in *C. longa* L. are linear diarylheptanoids that share similar size and structural characteristics with plant diterpenes, and are recognized as well-known secondary metabolites of Curcuma plants. Motivated by these findings, we further investigated the JHD activities of three commercially available curcuminoids, curcumin, DMC, and BDMC, which are constituents of *C. longa* L. (Figure 1). Interestingly, two of these curcuminoids, DMC and BDMC, exhibited robust interference with the heterodimer binding of Met–Tai, while showing no anti-yeast activity. These results highlight the potential of curcuminoids, particularly DMC and BDMC, as potent modulators of JHD activity, and shed light on their intricate role in interfering with juvenile hormone signaling pathways.

### 3.2. Changes in the Emergence Rates of D. melanogaster Larvae According to the Diet

LE3G, an isolated diterpene from the extract of *L. erythrocarpa*, has previously been reported to interfere with larval development in mosquitoes, fruit flies, and moths [29,31,32], and it was used as a positive control in our study. Building on this, we explored the impact of two commercially available curcuminoids, DMC and BDMC, on larval development in *D*. *melanogaster*. Similar to the larvicidal effects observed with LE3G, the introduction of DMC or BDMC into the diets of *D. melanogaster* larvae resulted in a concentration-dependent inhibition of larval development. Notably, diets containing 1% (*w*/*v*) DMC or BDMC extract exhibited a pronounced disruption in larval development, preventing progression to pupal stages (Figure 2). Furthermore, in diets supplemented with the JHD (LE3G, DMC, and BDMC at 1%, *w*/*v*), no adult flies were observed (Figure 3). This outcome suggests a significant impediment to the normal developmental transitions, emphasizing the potency of these compounds in disrupting the life cycle of *D. melanogaster*. These findings not only validate the larvicidal activity of LE3G, but also extend it to the curcuminoids DMC and BDMC.

### 3.3. DMC and BDMC Effects on JH-Dependent Gene Expression

*Krüppel homolog 1* (*Kr-h1*), identified as an early JH-inducible gene, assumes a pivotal role in insect metamorphosis by orchestrating the activation of the JH/Met/Tai complex [26]. In this study, we employed *Kr-h1* as a marker gene to elucidate its expression pattern in second-instar *D. melanogaster* larvae under the influence of various treatments. Upon treating *D. melanogaster* larvae with LE3G, DMC, and BDMC, a consistent trend emerged, revealing a decrease in *Kr-h1* expression compared with the control group (Figure 4). This downregulation suggests a potential inhibitory effect of these compounds on the JH/Met/Tai complex, aligning with their observed interference with larval development. In contrast, methoprene-treated flies exhibited an increase in *Kr-h1* expression (Figure 4). Methoprene, a JH analog, is known to mimic the effects of JH in insects, promoting a response akin to natural JH induction. This elevation in *Kr-h1* expression in methoprene-treated flies supports the expected activation of the JH signaling pathway. These findings underscore the modulatory impact of LE3G, DMC, and BDMC on *Kr-h1* expression, indicative of their influence on the JH-regulated processes crucial for insect metamorphosis.

## 4. Discussion

Secondary metabolites in plants are produced as defense mechanisms against insects and other herbivores, and have been used in this way throughout their evolutionary history [33,34]. Such secondary metabolites have become a research focus regarding the development of eco-friendly pesticides [35,36]. Plant-derived pesticides are considered ideal defense technologies, owing to their perceived advantages including low toxicity, targeted effectiveness against specific pests, and minimal environmental damage [37,38]. *C. longa* L., possessing insecticidal activity, has been utilized as an insecticide because of its diverse bioactive constituents that disrupt insect behavior and growth, making it possibly applicable for pest control in agriculture [39]. Curcuminoids, major components of *C. longa* rhizome powder, are known to show high insecticidal activity when used against various pests, such as *Culex pipiens* larvae [40]. Additionally, 45–60% growth inhibition and 10–15% mortality were confirmed as a result of an insect growth inhibition assay in *Schistocerca gregaria* (Forsskål, 1775) and *Dysdercus koenigii* (Fabricius, 1775), by injecting the nymphs with curcuminoids [41]. In our study, we investigated the impact of curcuminoids on *D. melanogaster* larvae. The findings revealed that larvae fed with a 1% curcuminoids diet failed to develop normally, indicating a pronounced inhibitory effect on larval and pupal development, particularly in relation to JH (Figure 3). This observation underscores the potential utility of curcuminoids as disruptors in the regulation of insect development, providing valuable insights for the development of novel and environmentally friendly pest control strategies.

Curcuminoids, the yellow polyphenols responsible for the biological activity of turmeric oleoresin [42], are linear diarylheptanoids. Chemically, they are composed of two benzene rings linked by unsaturated chains, and they are a mixture of curcumin [1,7-bis(4-hydroxy-3-methoxy-phenyl)-hepta-1,6-diene-3,5-dione] and two derivatives, DMC and BDMC [43]. Curcumin contains methoxy groups on both ends of the benzene rings, whereas DMC lacks one of these methoxy groups. Conversely, BDMC has no methoxy groups and possesses a simpler structure than curcumin and DMC. Considering this structural difference, it is inferred that DMC and BDMC show higher Met–Tai binding inhibitory activity compared to curcumin, which is likely due to the variation in the number of methoxy groups on both ends of the benzene rings (Figure 5). Furthermore, methoprene and pyriproxyfen have similar ring-shaped backbone structures (Figure 5) similar to JH [44]. The cyclopentenedione rings of curcuminoids and LE3G consist of two α,β-unsaturated ketone functional groups [45]. These compounds, referred to as curcuminoids, can interfere with the binding of the JH/Met/Tai complex by interacting with the methoxy and hydroxyl groups at both ends. However, elucidating the chemical binding with JH as a single determinant is challenging because the interaction with JH receptors is regulated by different structural features and chemical properties inherent to different compounds. Considering that curcuminoids possess phenolic rings, α- and β-unsaturated carbonyl groups, and multiple substituents, these distinctive structural features may significantly influence their interaction with JH receptors. The intricate interplay between the phenolic rings, unsaturated carbonyl groups, and substituents adds complexity to the understanding of how curcuminoids specifically modulate the JH/Met/Tai complex, and requires further investigation into the nuanced mechanisms at play in this molecular interaction.

In a previous study, LE3G, an extract of *L. erythrocarpa*, showed the opposite pattern of *Kr-h1* expression than that of JHD and methoprene [31]. JHD hinders larval development by interrupting heterodimer binding between Met and Tai through JH-mediating adjustments, whereas methoprene promotes heterodimer binding between Met and Tai by activating JH-mediating adjustments [46]. DMC and BDMC interfered with the development of *Drosophila* larvae (Figure 2), confirming that these substances had opposite effects compared with methoprene on *Kr-h1* expression (Figure 4). Additionally, it is believed that when insects ingest a diet containing a mixture of DMC or BDMC, it disrupts the regulation of JH-mediated genes, while also interfering with normal JH-dependent development. Therefore, the use of insecticidal methods using DMC or BDMC holds great promise for the formulation of novel pest control strategies. The appeal lies in their safety profile, as these substances biodegrade to non-toxic products. In addition, their potential suitability for integration into comprehensive pest management programs enhances their attractiveness as environmentally friendly alternatives. They are expected to serve as a model for the development of new synthetic analogs with favorable biological properties. Furthermore, as both curcuminoids are components of turmeric, they would be classified as substances harmless to humans and animals. As a result, they can enhance environmental protection and the value of agricultural products. By maintaining the sustainability of insecticide production and preserving a healthy environment, these plant-derived insecticides are considered potential candidates for new insecticidal substances.

## 5. Conclusions

This study highlights the insecticidal effects of curcuminoids, which are known to be the major constituents of *C. longa* L., particularly against *D. melanogaster* larvae. Components of the curcuminoids, DMC and BDMC, were shown to affect *D. melanogaster* larval and pupal development, suggesting that they interfere with JH-related processes. Structural differences among curcuminoids, particularly in the number of methoxy groups, likely contribute to variations in their binding affinity with the JH/Met/Tai complex. DMC and BDMC, whose effects on *Kr-h1* expression were found opposite to those of methoprene, disrupt the heterodimeric bond in Met–Tai, which is important for JH-mediated regulation. More importantly, the use of DMC or BDMC as an eco-friendly pesticide suggests that these substances can be presented as a safe alternative to chemical pesticides, with less environmental damage. In addition, the fact that these substances are classified as harmless to humans and animals and are environmentally friendly can contribute to environmental protection and enhancement of agricultural value. More research is required to determine the mechanisms through which DMC and BDMC affect larval and pupal development. Further investigations into the molecular mechanisms underlying this interference and the specific effects on insect physiology are also warranted to fully understand the implications for pest control strategies. The ability to use plant-derived insecticides to replace chemical insecticides could be valuable for environmentally friendly and sustainable agricultural practices.

## Figures and Tables

**Figure 1 insects-14-00959-f001:**
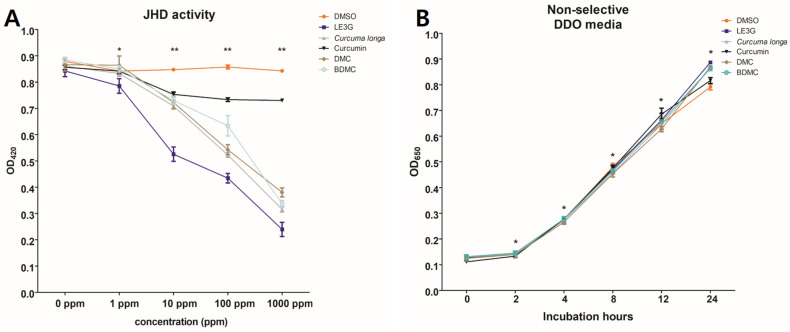
Juvenile hormone disruptor (JHD) activities of extracts from *Curcuma longa* L. and two curcuminoids, demethoxycurcumin (DMC) and bisdemethoxycurcumin (BDMC). (**A**) Met–Tai binding triggered by the addition of 0.1 ppm of JH III and the inhibition of the Met–Tai binding activity by treatment with 10 ppm of methyl lucidone (LE3G) was used as a positive control. The plant extract and compounds were added to the yeast culture at each shown concentration. * *p* < 0.05 and ** *p* < 0.001 (one-way ANOVA). (**B**) Anti-yeast activity tests of *C. longa* L. with JHD activity. Corresponding compounds were tested for their anti-yeast activity to investigate whether the reduced β–galactosidase activity resulted from JHD activity or anti-yeast toxicity. Mean values with associated error bars are presented as means ± standard deviation (*n* = 3) * *p* < 0.001 (one-way ANOVA).

**Figure 2 insects-14-00959-f002:**
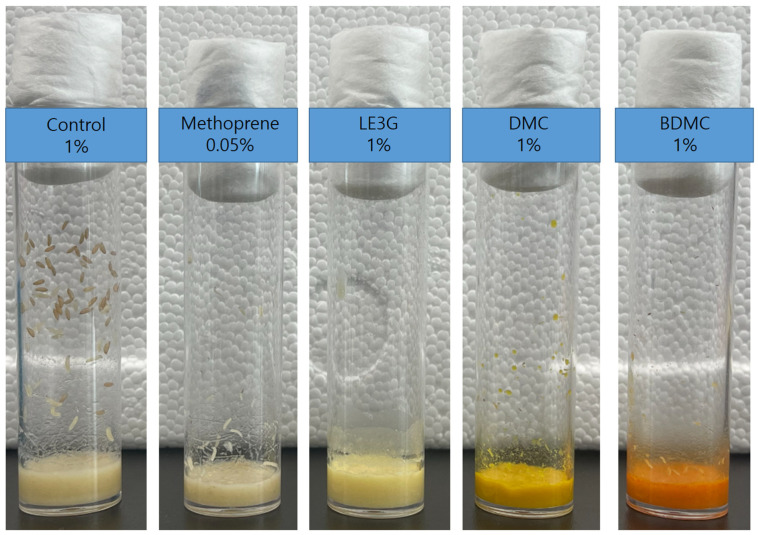
Effect of juvenile hormone analog (JHA) (methoprene) or juvenile hormone disruptor (JHD) (LE3G, DMC, and BDMC) compounds on *Drosophila melanogaster* larval development. LE3G: methyl lucidone; DMC: demethoxycurcumin; BDMC: bisdemethoxycurcumin.

**Figure 3 insects-14-00959-f003:**
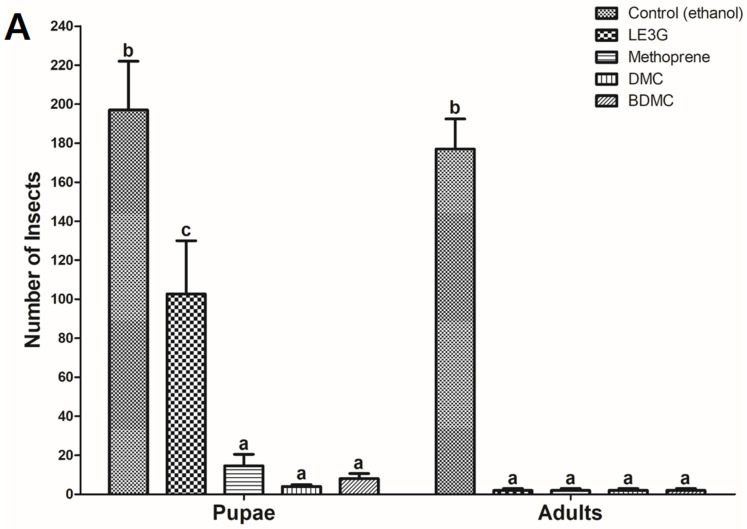
Effect of demethoxycurcumin (DMC) and bisdemethoxycurcumin (BDMC) on the growth and developmental of *Drosophila melanogaster* larvae. (**A**) Artificial diets were supplemented with either 1% (*w*/*v*) ethanol (control) or juvenile hormone disruptor (JHD) compounds (methyl lucidone (LE3G), DMC, and BDMC), or 0.05% (*w*/*v*) juvenile hormone analog (JHA) (methoprene). Mean values with associated error bars are presented as means ± standard deviation (*n* = 3). Different letters indicate significant differences at *p*  <  0.001 (one-way ANOVA followed by Tukey’s HSD test). (**B**) Dose–response curves for JHD effects on the non-development of *D. melanogaster* larvae into pupae (% of control). (**C**) Dose–response curves for JHD effects on the non-development of *D. melanogaster* pupae into adults (% of control).

**Figure 4 insects-14-00959-f004:**
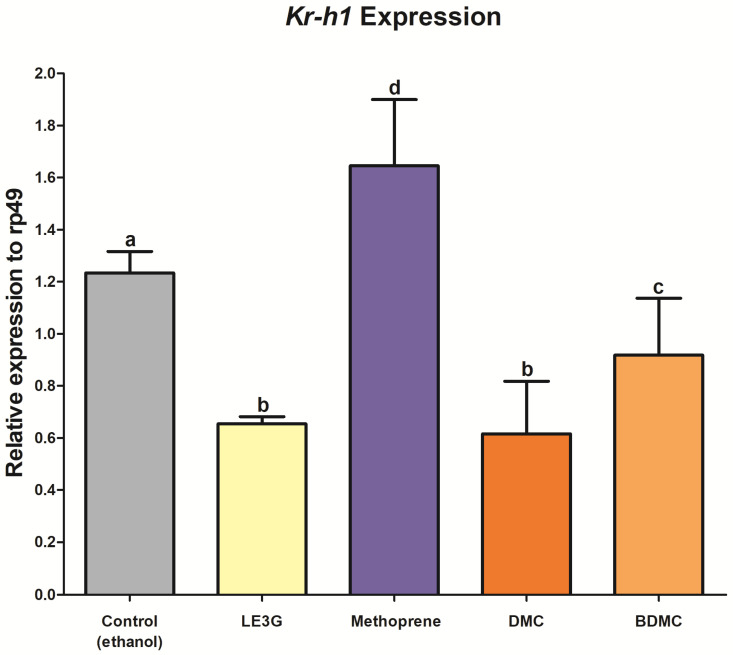
*Krüppel homolog 1* (*Kr-h1*) expression after demethoxycurcumin (DMC) or bisdemethoxycurcumin (BDMC) treatments. Total RNA was extracted from second-instar larvae that were fed a diet containing each component and analyzed using quantitative real-time polymerase chain reaction (qRT–PCR). Values and error bars indicate means ± SD (*n* = 3). Different letters indicate significant differences at *p* < 0.0001 (one-way ANOVA followed by Tukey’s HSD test).

**Figure 5 insects-14-00959-f005:**
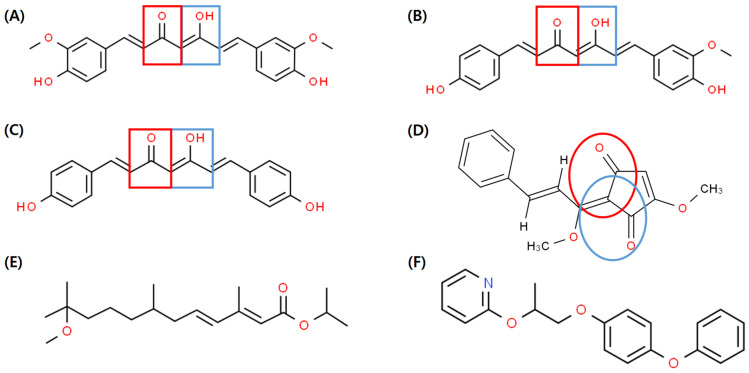
Chemical structures of juvenile hormone disruptor (JHD) and JH analog (JHa). JHD: (**A**) Curcumin, (**B**) DMC, (**C**) BDMC, (**D**) LE3G. JHa: (**E**) Methoprene, and (**F**) Pyriproxyfen. Red and blue shapes indicate α,β-unsaturated ketone moiety. DMC: demethoxycurcumin. BDMC: bisdemethoxycurcumin. LE3G: methyl lucidone.

**Table 1 insects-14-00959-t001:** Primer sequences used for qRT-PCR.

Gene	Primer
*Krüppel homolog 1*	Forward 5′-TCACACATCAAGAAGCCAACT-3′ Reverse 5′-GCTGGTTGGCGGAATAGTAA-3′
*Rp49*	Forward 5′-ATGCTAAGCTGTCGCACAAATG-3′ Reverse 5′-GTTCGATCCGTAACCGATGT-3′

**Table 2 insects-14-00959-t002:** Juvenile hormone disruptor (JHD) activity of plant extracts.

Species	Plant Part	Family	JHD Activity
*Curcuma longa* L.	Root	Zingiberaceae	1.217175
*Pulsatilla koreana* Nakai	Root	Ranunculaceae	1.101688
*Machilus thunbergii* Siebold & Zucc.	Trunk-bark	Lauraceae	1.056931
*Echinosophora koreensis* Nakai	Root	Fabaceae	0.960778
*Pinus densiflora* Siebold & Zucc.	Trunk-bark	Pinaceae	0.936298
*Alpinia officinarum*	Root	Zingiberaceae	0.904591
*Smilax sieboldii* Miq.	Leaf	Liliaceae	0.886733
*Scutellaria baicalensis*	Flower	Labiatae	0.825496
*Portulaca oleracea* L.	Whole	Portulacaceae	0.806894
*Magnolia kobus* DC	Leaf	Magnoliaceae	0.785417
*Broussonetia papyrifera* (L.) L’Hér. ex Vent.	Leaf	Moraceae	0.768187
*Syringa patula* (Palib.) Nakai	Leaf	Oleaceae	0.759335
*Agrimonia pilosa* Ledeb.	Whole	Rosaceae	0.750968
*Cudrania tricuspidata* (Carr.) Bureau ex Lavallée	Fruit	Moraceae	0.745534
*Psoralea corylifolia* (Babchi)	Seed	Fabaceae	0.658419
*Phlomis umbrosa* Turcz.	Whole	Labiatae	0.65772
*Cudrania tricuspidata* (Carr.) Bureau ex Lavallée	Trunk	Moraceae	0.613586
*Zingiber officinale*	Root	Zingiberaceae	0.546682
*Myristica fragrans*	Seed	Myristicaceae	0.535624
*Saururus chinensis* (Lour.) Baill.	Whole	Saururaceae	0.493735
*Glycyrrhiza uralensis*	Root	Fabaceae	0.469728
*Picrasma quassioides* (D. Don) Benn.	Trunk	Simaroubaceae	0.433324
*Sophora flavescens*	Root	Fabaceae	0.391224
*Morus alba* L.	Root	Moraceae	0.390836
*Actinostemma lobatum* Maxim.	Whole	Cucurbitaceae	0.381283
*Salvia miltiorrhiza* Bunge	Whole	Labiatae	0.363577
*Cudrania tricuspidata* (Carr.) Bureau ex Lavallée	Root	Moraceae	0.257315
*Eclipta prostrata*	Whole	Compositae	0.237714
*Magnolia obovata* Thunb	Trunk-bark	Lauraceae	0.176234
*Angelica keiskei*	Leaf	Apiaceae	0.134399

## Data Availability

The data presented in this study are available on request from the first author or corresponding author.

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
