# Peer review of "Disruptive Effects of Two Curcuminoids (Demethoxycurcumin and Bisdemethoxycurcumin) on the Larval Development of Drosophila melanogaster"

_insects, 2023, doi:10.3390/insects14120959_

Round 1

Reviewer 1 Report

Comments and Suggestions for Authors

In this article, the authors evaluated the insecticidal effects of curcuminoids, specifically demethoxycurcumin (DMC) and bisdemethoxycurcumin (BDMC), on Drosophila melanogaster larvae and pupae. Their findings indicate that these two compounds interfere with the juvenile hormone (JH)-related processes and affect insect growth and development. While the study provides valuable insights into the potential applications of DMC and BDMC as pesticides for managing insect pest populations, it is important to note that the article requires careful revision due to several misconceptions, improper data analyses, illogical conclusions, and writing issues.

Major comments:

1. The current title is misleading as the study lacks molecular characterization. A more accurate title could be "Disruptive Effects of Two Curcuminoids (Demethoxycurcumin and Bisdemethoxycurcumin) on the Larval Development of Drosophila melanogaster."

2. The abstract currently lacks a clear presentation of the main findings. It is recommended to emphasize the results more prominently, providing specific details such as the numerical reduction in larvae. Additionally, the key findings related to juvenile hormone (JH) should be highlighted.

3. In line 180, how were the authors sure that high JHD activity was observed in C. longa without any statistical analysis. It is advisable to perform appropriate statistical tests to demonstrate the significant differences in JHD activity. In Figure 1, 3, 4, the data should be analyzed by using one-way ANOVA followed by multi-comparison and incorporating lowercase letters to indicate significant differences among treatments.

4. Refine the conclusion to focus solely on the findings. There are no results for the conclusion “disrupt the heterodimeric bond between Met-taiman, which is important for JH-mediated regulation. The certainty about the safety and lack of harmful effects of the compounds needs to be substantiated with corresponding results. How are authors sure that these compounds are safe and have no harmful effects on environment and humans without results. More importantly, the use of DMC or BDMC as an eco-friendly pesticides? If no results are available, refrain from making definitive statements about the safety of the compounds for the environment and humans. Remove statements suggesting the substances as safe alternatives without supporting data.

Minor comments:

In Line 16, remove or

In Line 24, replace “and they exhibit insect growth inhibitory effects” and exhibit growth inhibitory effects on insects

In Lines 30, 40, consider to rewrite the sentence.

In Lines 75, 76, 77, 78, it is hard to understand.

In Lines 108, 176, 180, 198, 210, 213, 219, 236, 237, pay attention to species names, which should be italic and abbreviated; the compound names should be abbreviated, and there is no need to write full compound name as you already abbreviated them in the beginning.

Comments on the Quality of English Language

Moderate editing of English language is required.

Author Response

Question: The current title is misleading as the study lacks molecular characterization. A more accurate title could be "Disruptive Effects of Two Curcuminoids (Demethoxycurcumin and Bisdemethoxycurcumin) on the Larval Development of Drosophila melanogaster."

Answer: We appreciate the reviewer’s comment. We've made the changes you mentioned. Thank you for the opportunity to correct the mistake according to the reviewer's point.

  • Disruptive Effects of Two Curcuminoids (Demethoxycurcu-min and Bisdemethoxycurcumin) on the Larval Development of Drosophila melanogaster

Question: The abstract currently lacks a clear presentation of the main findings. It is recommended to emphasize the results more prominently, providing specific details such as the numerical reduction in larvae. Additionally, the key findings related to juvenile hormone (JH) should be highlighted.

Answer: We appreciate the reviewer’s comment. We've made the changes you mentioned. Thank you for the opportunity to correct the mistake according to the reviewer's point.

  • Juvenile hormones (JHs) play a central role in insect development, reproduction, and various physiological functions. Curcuminoids generally exhibit a wide range of biological activities, such as antioxidant, anti-inflammatory, antibacterial, and insecticidal, and they exhibit insect growth inhibitory effects. However, research on insecticidal properties of curcuminoids has been limited. Moreover, to the best of our knowledge, studies on JHs of insects and curcuminoids are lacking. Therefore, this study aimed to identify the substances that act as JH disruptors (JHDs) from edible plants. Demethoxycurcumin (DMC) and bisdemethoxycurcumin (BDMC), two curcuminoids from the turmeric plant Curcuma longa L. inhibited the formation of a methoprene-tolerant (Met)–Taiman (Tai) heterodimer complex in Drosophila melanogaster, as shown through in vitro yeast-two-hybrid assays. An artificial diet containing 1% (w/v) DMC or BDMC significantly reduced the number of D. melanogaster larvae in a concentration-dependent manner; larval development was disrupted, preventing the progression of larvae to pupal stages, resulting in an absence of adults. Building on the results obtained in this study on curcuminoids, researchers can use our study as a reference to develop eco-friendly pesticides.

Question: In line 180, how were the authors sure that high JHD activity was observed in C. longa without any statistical analysis. It is advisable to perform appropriate statistical tests to demonstrate the significant differences in JHD activity. In Figure 1, 3, 4, the data should be analyzed by using one-way ANOVA followed by multi-comparison and incorporating lowercase letters to indicate significant differences among treatments.

Answer: We appreciate the reviewer’s comment. We've made the changes you mentioned. Thank you for the opportunity to correct the mistake according to the reviewer's point.

  • Statistical analysis was performed using one way ANOVA for Figures 1, 3, and 4, and significant differences are indicated using lowercase letters.

Question: Refine the conclusion to focus solely on the findings. There are no results for the conclusion “disrupt the heterodimeric bond between Met-taiman, which is important for JH-mediated regulation”. The certainty about the safety and lack of harmful effects of the compounds needs to be substantiated with corresponding results. How are authors sure that these compounds are safe and have no harmful effects on environment and humans without results. More importantly, the use of DMC or BDMC as an eco-friendly pesticides? If no results are available, refrain from making definitive statements about the safety of the compounds for the environment and humans. Remove statements suggesting the substances as safe alternatives without supporting data.

Answer: We appreciate the reviewer’s comment.

  • The decrease in beta-galactosidase values with increasing concentrations of DMC and BDMC (Figure 1-A) suggests interference with the binding of the heterodimer. This was observed when DMC or BDMC was introduced, even in the presence of JH, into the Y187 strain transformed with the Met/Taiman complex, leading to a reduction in activity (starting from approximately 10 ppm). These results indicate that DMC and BDMC hinder the interaction between Met and Taiman. Furthermore, consistent with previous studies using the same experimental approach, various plant extracts were found to disrupt the binding of Met/Taiman and Met/SRC complexes (Ref: Identification of plant compounds that disrupt the insect juvenile hormone receptor complex, A plant diterpene counteracts juvenile hormone-mediated gene regulation during Drosophila melanogaster larval development, Species-Specific Interactions between Plant Metabolites and Insect Juvenile Hormone Receptors).

  • Additionally, an anti-yeast test was conducted on cells (Figure 1-B), confirming that there was no observable impact on the cells. Recent studies have shown that DMC and BDMC exhibit potential in inducing apoptosis in various cancer cells, as well as revealing the therapeutic potential of curcumin in diseases. Findings such as "Curcumin, demethoxycurcumin, and bisdemethoxycurcumin induced caspase-dependent and –independent apoptosis in HOS cells" underscore the cytotoxic effects against cancer cells (Curcumin, demethoxycurcumin, and bisdemethoxycurcumin induced caspase-dependent and –independent apoptosis via Smad or Akt signaling pathways in HOS cells, Dimethoxycurcumin reduces proliferation and induces apoptosis in renal tumor cells more efficiently than demethoxycurcumin and curcumin, Multiple health benefits of curcumin and its therapeutic potential). Considering these results, it can be inferred that DMC and BDMC do not exert harmful effects on humans.

  • Based on the results, it is believed that DMC and BDMC could serve as environmentally friendly insecticides in the future. The confirmation that these compounds do not have adverse effects on cells suggests that, when used as additives or independently, DMC and BDMC may function as safe and environmentally friendly insecticides, similar to conventional insecticides.

Minor comments:

Question: In Line 16, remove “or”

Answer: We appreciate the reviewer’s comment. We've made the changes you mentioned. Thank you for the opportunity to correct the mistake according to the reviewer's point.

Question: In Line 24, replace “and they exhibit insect growth inhibitory effects” and exhibit growth inhibitory effects on insects

Answer: We appreciate the reviewer’s comment. We've made the changes you mentioned. Thank you for the opportunity to correct the mistake according to the reviewer's point. I also had my English proofread by a professional.

Question: In Lines 30, 40, consider to rewrite the sentence.

Answer: We appreciate the reviewer’s comment.

  • Based on the results, it is believed that DMC and BDMC could serve as environmentally friendly insecticides in the future. The confirmation that these compounds do not have adverse effects on cells suggests that, when used as additives or independently, DMC and BDMC may function as safe and environmentally friendly insecticides, similar to conventional insecticides.

Question: In Lines 75, 76, 77, 78, it is hard to understand.

Answer: We appreciate the reviewer’s comment. We've made the changes you mentioned. Thank you for the opportunity to correct the mistake according to the reviewer's point.

  • JHs act through a receptor complex consisting of Met and steroid receptor coactivator (SRC; Taiman in melanogaster or βFtz-F1 Interacting Steroid Receptor Coactivator FISC in A. aegypti) to induce the transcription of specific genes. Met heterodimerizes with an SRC and binds to JH with high affinity, activates transcription.

Question: In Lines 108, 176, 180, 198, 210, 213, 219, 236, 237, pay attention to species names, which should be italic and abbreviated; the compound names should be abbreviated, and there is no need to write full compound name as you already abbreviated them in the beginning.

Answer: We appreciate the reviewer’s comment. We've made the changes you mentioned. Thank you for the opportunity to correct the mistake according to the reviewer's point.

Question: Moderate editing of English language is required.

Answer: I had my entire paper proofread by an English language expert.

Reviewer 2 Report

Comments and Suggestions for Authors

This paper first assays a number of edible plants for disruption of the Met-Taiman interaction in a previously validated yeast two hybrid assay.  They find that extracts of the tumeric plant Curcuma longa and two curcuminoids, demethoxycurcumin (DMC) and bisdemethoxycurcumin (BDMC), had high activity in blocking the Met-Taiman interaction.   When 1% DMC or BDMC was fed to larvae, only a few pupariated and none formed adults.  These compounds also decreased the amount of Kruppel homolog mRNA present in second instar larvae which is thought to be due to a decreased titer of juvenile hormone.

              In general, the study appears to be carefully done.  However, I have two concerns:

1)      In the experiments on Kr-h1 expression, were you selecting for the slow growing larvae?  At 25° C, Drosophila larvae should be molting to the third instar on day 3 after egg laying.  Please explain.

2)      How did you determine the 1% concentration to use in the dietary experiments?  A dose-response experiment is needed.

Minor corrections necessary:

1)      All genus and species names and gene names need to be italicized throughout. You have a mixture.

2)      Line 16: should be “in vivo”.

3)      Line 78: Met binds…

4)      Line 234: Kr-h1

5)      Fig. 4: The label should be “Kr-h1 Expression”.

Author Response

Question: In the experiments on Kr-h1 expression, were you selecting for the slow growing larvae? At 25° C, Drosophila larvae should be molting to the third instar on day 3 after egg laying. Please explain

Answer: We appreciate the reviewer’s comment.

We conducted experiments by selecting second-instar larvae from vials three days after oviposition, before they developed into third instars. To ensure consistent data, we selected second-instar larvae of uniform size for our experiments.

Question: How did you determine the 1% concentration to use in the dietary experiments? A dose-response experiment is needed.

Answer: We appreciate the reviewer’s comment. In the body, I wrote the following

Each JHD (LE3G, DMC, BDMC) was first dissolved in ethanol at a concentration of 20 mg/ml (100%). The 100% solution was then further diluted with ethanol to produce a 10-fold solution of the corresponding JHD concentration. For example, to formulate a diet containing 1% JHD, 1 volume of the 10% stock solution was mixed with 9 volumes of Drosophila diet, resulting in a 1% solution.

Minor corrections necessary:

Question: 1) All genus and species names and gene names need to be italicized throughout. You have a mixture.

Answer: We appreciate the reviewer’s comment. I changed all of the gene names to italics. Thank you for the opportunity to correct the mistake according to the reviewer's point.

Question: 2) Line 16: should be “in vivo”.

Answer: : We appreciate the reviewer’s comment. I think in vitro is correct in that sentence. Thanks for the advice. 

Question: 3) Line 78: Met binds…

Answer: We appreciate the reviewer’s comment. We changed it to Met. Thank you for the opportunity to correct the mistake according to the reviewer's point.

Question: 4) Line 234: Kr-h1

Answer: We appreciate the reviewer’s comment. We changed it to Kr-h1. Thank you for the opportunity to correct the mistake according to the reviewer's point.

Question: 5) Fig. 4: The label should be “Kr-h1 Expression”.

Answer: We appreciate the reviewer’s comment. We changed it to Kr-h1 Expression. Thank you for the opportunity to correct the mistake according to the reviewer's point.

Reviewer 3 Report

Comments and Suggestions for Authors

See attach

Comments on the Quality of English Language

Quality of the English language is fine although some minor typographical errors should be corrected

Author Response

Question: L14, I think it is more appropriate to say here ‘Several plant species have diterpene compounds with juvenile hormone (JH) disruption activity in insects’. Use ‘disruptor’ in expressions like ‘substances that act as JH disruptors…’ like in L22.

Answer: We appreciate the reviewer’s comment. The existing literature we've studied, as reflected in works such as 'A plant diterpene counteracts juvenile hormone-mediated gene regulation during Drosophila melanogaster larval development' and 'Inducible Expression of Several Drosophila melanogaster Genes Encoding Juvenile Hormone Binding Proteins by a Plant Diterpene Secondary Metabolite, Methyl Lucidone,' seems to align with the terminology we've employed here. We'll continue using the term 'disruptors' in accordance with the literature.

Thank you for the opportunity to correct the mistake according to the reviewer's point.

Question: L15, consider saying ‘interfering with the JH-mediated formation’.

Answer: We appreciate the reviewer’s comment. We've made the changes you mentioned. Thank you for the opportunity to correct the mistake according to the reviewer's point.

Question: L41-42, change the order in the sentence to say ‘Plants use secondary metabolites including nicotine, rotenone, lianas, sabadala, pyrethrum, neem, and turmeric, as natural defenses against insect pests [3]’.

Answer: We appreciate the reviewer’s comment. We've made the changes you mentioned. Thank you for the opportunity to correct the mistake according to the reviewer's point.

Question: L52-53, consider saying ‘Research studies on insecticidal properties have explored the efficacy of curcuminoids…’.

Answer: We appreciate the reviewer’s comment. We've made the changes you mentioned. Thank you for the opportunity to correct the mistake according to the reviewer's point.

Question: L54, always put insect and plant names (in full or abbreviated) in italics, like Helicoverpa armigera here, D. melanogaster in L108, 120, Curcuma longa in L181, etc. Insert a full stop after [15].

Answer: We appreciate the reviewer’s comment. We've made the changes you mentioned. Thank you for the opportunity to correct the mistake according to the reviewer's point.

Question: L54-55, in the next sentence ‘In particular, curcuminoids I, II, III inhibited the P-glycoprotein ATPase in H. armigera [16] and insecticidal activity against Aedes aegypti [17]’, it is unclear whether the effect of these curcuminoids on Aedes aegypti is just ‘insecticidal activity’ or ‘inhibition of the insecticidal activity’. If it is ‘insecticidal activity’ it should say ‘and displayed insecticidal activity…’.

Answer: We appreciate the reviewer’s comment. We've made the changes you mentioned. Thank you for the opportunity to correct the mistake according to the reviewer's point. We have changed to displayed insecticidal activity.

Question: L59, please correct to say ‘It was also found…’.

Answer: We appreciate the reviewer’s comment. We've made the changes you mentioned. Thank you for the opportunity to correct the mistake according to the reviewer's point.

Question: L64, insert ‘so far’ at the beginning of the sentence to say ‘to our knowledge, so far studies on the juvenile hormones (JHs) … have not been conducted’.

Answer: We appreciate the reviewer’s comment. We've made the changes you mentioned. Thank you for the opportunity to correct the mistake according to the reviewer's point.

Question: L66, insert comma after ‘allatum’.

Answer: We appreciate the reviewer’s comment. We've made the changes you mentioned. Thank you for the opportunity to correct the mistake according to the reviewer's point.

Question: L70-71, remove ‘the’ and ‘in D. melanogaster’ to say ‘as a JH receptor in mutant studies of D. melanogaster, accelerating research on JH signaling [23]’.

Answer: We appreciate the reviewer’s comment. We've made the changes you mentioned. Thank you for the opportunity to correct the mistake according to the reviewer's point.

Question: L83, remove ‘studied’.

Answer: We appreciate the reviewer’s comment. We've made the changes you mentioned. Thank you for the opportunity to correct the mistake according to the reviewer's point.

Question: L103, concentration of the stock solution?

Answer: We appreciate the reviewer’s comment. We've made the changes you mentioned (Each reagent was prepared as a stock solution (20mg/ml) in either dimethyl sulfoxide (DMSO) or methanol). Thank you for the opportunity to correct the mistake according to the reviewer's point.

Question: L128, it should be also stated the amount of JHIII in the OD420Control.

Answer: We appreciate the reviewer’s comment. We've made the changes you mentioned (· OD420 Control is the absorbance of yeast cells treated with JH III at 0.1ppm.). Thank you for the opportunity to correct the mistake according to the reviewer's point.

Question: L181, make sure to abbreviate the plant names once they have been cited for the first time, like C. longa. Also, in L186.

Table 2. It is noteworthy the work done by the authors testing the JH disruption activity of 30 plants, but since only one (Curcuma longa L.) was finally selected for further study, I recommend to move Table 2 to the Supplementary Material. Also, in title use ‘disruption’ instead of ‘disrupter’.

Answer: We appreciate the reviewer’s comment. We've made the changes you mentioned (L181, 186). Thank you for the opportunity to correct the mistake according to the reviewer's point.

The plants extracts activity in Table 2 are considered important, and it is recommended to include their details in the main text. Also, As mentioned earlier, based on the existing literature we studied, we believe that the title of Table 2. should be disruptors.

Question: Figure 1. Lettering is too small.

Answer: We appreciate the reviewer’s comment. We've made the changes you mentioned. Thank you for the opportunity to correct the mistake according to the reviewer's point.

Question: L209, remove ‘this substance was’ (unnecessary).

Answer: We appreciate the reviewer’s comment. We've made the changes you mentioned. Thank you for the opportunity to correct the mistake according to the reviewer's point. I had my entire paper proofread by an English language expert.

Question: L234, insert a reference after ‘JH/Met/Tai complex’.

Answer: We appreciate the reviewer’s comment. We've made the changes you mentioned. Thank you for the opportunity to correct the mistake according to the reviewer's point. Added the Kayukawa et al. 2012 reference.